# Placebo Effect of Caffeine on Maximal Strength and Strength Endurance in Healthy Recreationally Trained Women Habituated to Caffeine

**DOI:** 10.3390/nu12123813

**Published:** 2020-12-13

**Authors:** Aleksandra Filip-Stachnik, Michal Krzysztofik, Magdalena Kaszuba, Agata Leońska-Duniec, Wojciech Czarny, Juan Del Coso, Michal Wilk

**Affiliations:** 1Institute of Sport Sciences, The Jerzy Kukuczka Academy of Physical Education in Katowice, 40-065 Katowice, Poland; a.filip@awf.katowice.pl (A.F.-S.); m.krzysztofik@awf.katowice.pl (M.K.); m.kaszuba1@interia.pl (M.K.); 2Faculty of Physical Education, Gdansk University of Physical Education and Sport, 80-336 Gdansk, Poland; leonska.duniec@gmail.com; 3College of Medical Sciences, Institute of Physical Culture Studies, University of Rzeszow, 35-959 Rzeszow, Poland; wojciechczarny@wp.pl; 4Centre for Sport Studies, Rey Juan Carlos University, 28942 Fuenlabrada, Spain; juan.delcoso@urjc.es

**Keywords:** resistance exercise, muscle performance, belief effects, psychological advantages, ergogenic aids

## Abstract

Background: By using deceptive experimental designs, several investigations have observed that trained individuals may increase their performance when told they were given caffeine, when in fact they received a placebo (i.e., the placebo effect of caffeine). However, most of these investigations on the placebo effect of caffeine used individuals with low caffeine consumption or did not report habitual caffeine consumption, especially in studies analyzing resistance-based exercise. Hence, it is unknown if habitual caffeine consumers benefit from the placebo effect of caffeine on exercise performance. Thus, the aim of the present study was to analyze the placebo effect of caffeine on maximal strength and strength-endurance performance during the bench press exercise (BP) in women with mild–moderate daily consumption of caffeine. Methods: Thirteen resistance-trained women (BP one-repetition maximum (1RM) = 40.0 ± 9.7 kg) habituated to caffeine (4.1 ± 1.7 mg/kg/day) completed a deceptive randomized experimental design with two experimental trials. On one occasion, participants were told that they would receive 6 mg/kg of caffeine but received a placebo (PLAC), and on other occasions, participants did not receive any substance and were told that this was a control situation (CONT). In each experimental trial, participants underwent a 1RM BP test and a strength-endurance test consisting of performing the maximal number of repetitions at 50% of their 1RM. Results: In comparison to CONT, PLAC did not enhance 1RM (40.0 ± 10.5 kg vs. 41.0 ± 9.5 kg, respectively; *p* = 0.10), nor did it enhance the number of repetitions (32.2 ± 5.1 vs. 31.8 ± 4.5; *p* = 0.66) or mean power (130 ± 34 vs. 121 ± 26; *p* = 0.08) in the strength-endurance test. Conclusion: Informing participants that they were given caffeine, when in fact they received a placebo, did not modify any performance variable measured in this investigation. Thus, the use of the placebo effect of caffeine seemed an ineffective strategy to enhance muscle strength and strength endurance during the BP exercise in women with mild–moderate consumption of caffeine.

## 1. Introduction

Caffeine (1,3,7-trimethylxanthine) is one of the most frequently used supplements before training and competition, with the aim of enhancing performance and readiness to exercise [1]. In fact, three out of four elite athletes were reported to have caffeine present in their urine collections for doping control, indicating the wide utilization of caffeine before or during competition [2]. Furthermore, urinary caffeine concentration has increased since the removal of caffeine from the list of banned substances of the World Antidoping Agency in 2004, suggesting an increasing use of this stimulant in elite sport [3]. In the last years, a number of studies have supported the performance-enhancing effects of caffeine across a wide range of sporting activities, including aerobic exercise [4], anaerobic-like exercise [5], and resistance exercise [6,7,8]. Overall, there is ample consensus to consider the antagonistic role of caffeine and its two metabolites, paraxanthine and theophylline, in adenosine receptors, as the main mechanism behind its ergogenic effects during exercise. Caffeine can bind adenosine A_1_, A_2A_, and A_2B_ receptors in the central nervous system, reducing the fatiguing effect of adenosine [9]. However, several investigations have suggested that the belief of having ingested caffeine may also increase performance [10,11], suggesting that psychological factors may also contribute as a mechanism of action when caffeine is consumed in sporting settings. In this regard, the increase in performance produced just by the perception of consuming caffeine, without actually consuming it, is often referred to as a placebo effect of caffeine which is associated with expectancy of oral caffeine consumption, as in most cases it is perceived as positive to enhance performance [12].

The placebo effect is not a phenomenon unique for caffeine, as the change in motivation and self-efficacy just by the expectation of ingesting an active substance has been widely reported in the literature [13]. By using deceptive experimental designs, it has been found that a placebo can increase performance when participants are informed that the substance ingested is caffeine [14,15,16]. In previous investigations on this topic, the deceptive designs employed entailed different approaches to induce expectancy of caffeine consumption. In general, expectancy was induced by telling individuals that they had received an ergogenic dose of caffeine but administering an inert substance instead. Habitually, the performance outcomes of this protocol are compared to a control situation with no substance administration or by using a placebo-controlled situation in which participants who receive a placebo are told that they have received a placebo. However, the most complex deceptive protocols include four trials with a combination of information for the participants about the received substances and changes in the substances administered (i.e., informed caffeine/received caffeine, informed caffeine/received placebo, informed placebo/received caffeine, and informed placebo/received placebo). To date, most of the research has confirmed the placebo effect of caffeine on several forms of exercise [10,11,14,15,16,17,18,19,20], irrespective of the protocol used to induce the placebo effect of caffeine. In some of these investigations, participants further increased performance when they knowingly ingested caffeine, and the belief that they were ingesting caffeine seemed to be necessary for obtaining an ergogenic effect [10,21]. Interestingly, in most of these investigations, participants were naïve, or low caffeine consumers, while the only investigation that failed to report a placebo effect of caffeine used individuals habituated to this substance [16,22].

The main factor determining the placebo effect of caffeine in the above-mentioned studies is related to the expectation of receiving caffeine, as it is recognized as a potentially beneficial ergogenic aid [23]. However, it has to be emphasized that chronic caffeine intake may modify the acute effect of caffeine on sport performance [24], so that individuals habituated to caffeine obtain no, or lesser, benefits from acute caffeine intake [8,25,26,27]. Thus, the potential placebo effect of caffeine may also depend on the daily habitual consumption of caffeine of the participants [28] as the deceptive protocol may be unsuccessful in those individuals who are well familiarized with the feelings of acute caffeine intake. Therefore, the main goal of this study was to assess the placebo effect of caffeine on maximal strength and strength endurance in women habituated to caffeine. We hypothesized that the placebo effect of caffeine would not be present in recreationally trained women habituated to caffeine due to experience related to habitual caffeine intake.

## 2. Materials and Methods

### 2.1. Experimental Design

A deceptive, randomized, and cross-over design was used for this investigation. All testing was performed at the Strength and Power Laboratory of the Academy of Physical Education in Katowice under controlled ambient conditions. All experiments took place at the same time of day to avoid circadian variation. Each participant performed two familiarization sessions, one session to assess their one-repetition maximum (1RM), and two experimental trials, for a total of five different visits to the laboratory, separated by a 1-week interval. For the two experimental trials, participants were randomly assigned to two conditions: (1) ingestion of a capsule containing a placebo (PLAC), but participants were told that they were given 6 mg of caffeine per kg of body mass, or (2) not ingesting any capsule (CONT), while participants were told that this was a control situation to assess the effect of caffeine on muscle performance. In the PLAC condition, participants received a capsule containing all-purpose flour, 1 h before the onset of the exercise protocol. In the CONT condition, participants did not receive any treatment to increase the belief that they were ingesting caffeine in the PLAC condition. The exercise protocol included a measurement of their 1RM in the bench press (BP) exercise and a strength-endurance test consisting of performing the highest number of repetitions against a load representing 50% of their 1RM, as measured in the prior test. The study protocol was approved by the Bioethics Committee for Scientific Research at the Academy of Physical Education in Katowice, Poland (3/2019), according to the ethical standards laid down in the 1964 Declaration of Helsinki and its later amendments. All participants provided their written informed consent prior to participation in this study.

### 2.2. Study Participants

Power analysis indicated that a minimum sample size of 12 participants should be included in the study in order to detect an effect size (ES) of 0.89, obtained from a previous study examining placebo effects of caffeine on the number of repetitions performed to volitional failure [10]. Power analysis was performed using the following parameters: type of analysis was set for paired sample *t*-test, the required power was set to 0.80, and alpha was set to 0.05 (G*Power software, v.3.1.9.2). Based on a power analysis, we recruited 13 healthy recreationally trained women into the study (Table 1). The inclusion criteria were as follows: (a) free from neuromuscular and musculoskeletal disorders; (b) habitual caffeine user within the range of mild/moderate consumption, as per previous proposed thresholds for classifying individuals in sport performance research according to their habitual caffeine consumption [29]; (c) resistance-trained; (d) no medication or dietary supplement usage within the 3 previous months (oral contraceptives were allowed); and (e) a self-described satisfactory health status. Eleven women were regularly taking oral contraceptives for at least 3 months prior to the investigation. Participants were excluded if they reported (a) a positive smoking status or (b) a potential allergy to caffeine. Habitual caffeine intake was measured by using a modified version of the validated questionnaire by Bühler et al. [30] that recorded the type and amount of caffeine-containing foods and dietary supplements, and this information was obtained for the 4 weeks before the start of the experiment, following previous recommendations [29]. The participants were instructed to maintain their usual training, hydration, and dietary habits during the study period (with the exception of refraining from any source of dietary caffeine for 12 h prior to trials). In addition, participants registered their food intake using “MyFitnessPal” software [31] 24 h before the testing procedures. Last, participants were asked to avoid strenuous exercise for 24 h prior to each trial.

### 2.3. Familiarization Session and One-Repetition Maximum Test

Since the 1RM test and the sets until muscular failure were not a usual practice for participants, 3 weeks before the onset of the experimental trials, two familiarization sessions separated by 1 week were performed, in order to restrict learning effects during the experiment and decrease the risk of injury [32]. In each familiarization session, participants arrived at the laboratory at the same time of day and performed a standardized warm-up. Then, participants performed one set of the BP exercise consisting of the maximal number of repetitions at a load of approximately 50% of their 1RM. One week before onset of the experimental trials, participants underwent a 1RM BP testing according to previous guidelines [33]. In the last familiarization session, participants were informed about the potential ergogenic effects of acute caffeine intake on maximal strength and on muscular strength-endurance. The expectancy of positive effects of caffeine on performance was induced by informal but structured and normalized conversations with participants provided by a qualified sport dietitian. Specifically, participants were shown a published infographic on caffeine and resistance exercise [34], and the dietitian analyzed benefits for sport and exercise performance and explained the guidelines for caffeine supplementation.

### 2.4. Experimental Protocol

Participants arrived at the laboratory between 9:00 and 11:00 a.m. Upon arrival, participants received the treatment assigned for the session (i.e., PLAC or CONT) and rested for 45 min. Then, participants performed a warm-up that included 10 min of cycling and a single set of 8 reps of 50% 1RM, 6 reps of 70% 1RM, and 3 reps of 80% 1RM, established during the familiarization. The resting interval between the sets used for warming-up was 3 min. Then, participants performed the 1RM test to assess upper-body maximal muscle strength. The initial load was set at 90% of the estimated 1RM; in each subsequent attempt, the load was increased by between 1.25 and 5 kg until the load that represented the 1RM was reached. The 1RM load was determined using a maximum of two to five attempts with a resting interval of 5 min between attempts. After 5 min rest, the strength-endurance test was performed until muscular failure with the 50% of 1RM load, measured in the previous test. Muscular failure was defined as the inability to perform another concentric movement in its entire range of motion [35]. Participants were told that the concentric and eccentric phase of each repetition during the strength-endurance test should be performed at their maximal possible velocity. A linear position transducer system (Tendo Power Analyzer, Tendo Sport Machines, Trencin, Slovakia) was used to assess bar velocity during each repetition of the strength-endurance test [36], and mean and peak velocities were obtained for the test. The number of repetitions and the time under tension were also recorded. Execution technique during the strength-endurance test was monitored by two experienced researchers who gave a standardized verbal encouragement to aid in the obtaining of the highest velocity in each repetition. 

### 2.5. Statistical Analysis

The Shapiro–Wilk test was used in order to verify the normality of the data. The differences between the PLAC vs. CONT was identified using paired T-tests. The relative PLAC–CONT effect was also calculated as the difference between trials in percentage and through effect sizes (Cohen’s *d*). The magnitude of the effect size was interpreted as follows: large, *d* > 0.8; moderate, *d* between 0.8 and 0.5; small, *d* between 0.49 and 0.20; and trivial, *d* < 0.2 [37]. Statistical significance was set at *p* < 0.05. All statistical analyses were performed using Statistica 9.1. Data are presented as means ± standard deviations.

## 3. Results

The Shapiro–Wilk test showed normal data distribution for all measure variables. In comparison to CONT, PLAC did not change the load achieved in the 1RM test (Table 2). Furthermore, PLAC did not produce any statistically significant effect in the number of repetitions, the time under tension, or in the power-related variables during the strength-endurance test (SET). Further, a moderate effect size was observed in peak velocity between the PLAC and CONT conditions (Table 2).

## 4. Discussion

The main finding of the study was that there was no placebo effect of caffeine in women habituated to caffeine when misled to believe that they had ingested 6 mg/kg of caffeine. The lack of placebo effect of caffeine was evident on all muscle performance variables measured during the protocol used in the present study, based on the measurement of bench press maximal strength and strength endurance. Therefore, the use of a deceptive protocol to induce the placebo effect of caffeine is likely an ineffective measure to enhance muscle performance in recreationally trained women with habitual mild–moderate daily consumption of caffeine. 

To the best of our knowledge, this is the first study analyzing the placebo effect of caffeine on exercise performance in women habituated to this stimulant. In contrast to the outcomes of this study, previous research on male participants showed a placebo effect of caffeine on resistance exercise performance [11,15,18]. Pollo et al. [15] showed that placebo intake increased exercise performance during a strength-endurance test in a group of men who believed they had ingested high doses of caffeine. Similarly, a placebo effect of caffeine was obtained by Duncan et al. [11], who additionally strengthened their deceptive experimental design by providing the participants with scientific data confirming the effectiveness of acute caffeine intake on resistance exercise. Lastly, Costa et al. [18] observed that placebo intake, when the athletes were informed they were taking caffeine, was effective to enhance bench press throw performance in Paralympic weightlifters. The main difference between these three investigations and the current protocol, beyond the sex of the participants, is the previous habituation to caffeine. In the current investigation, participants were habituated to caffeine because they reported ingestion, on average, of 4.1 ± 1.6 mg of caffeine per kg of body mass per day. Due to a higher daily caffeine intake, beliefs of acute caffeine intake in habitual users of caffeine may differ in comparison to naive or low users. It has been found that chronic caffeine ingestion results in more newly created adenosine receptors that partially reduce the blocking action of caffeine on the central nervous system [38] and may modify the physiological and cognitive responses to acute caffeine intake, which could negatively impact its ergogenic effect [39,40]. If such suppressed reactions represent beliefs of habitual caffeine users, it could explain the lack of placebo effect of caffeine in this group of habitual users of caffeine. Other potential reasons for the lack of a placebo effect of caffeine in habitual users of caffeine may be associated with the higher expectancy of caffeine effectiveness to increase performance, as this may be one of the reasons why they became users of this stimulant. In this regard, it is probable that habitual users of caffeine are more prone to detect when they have actually ingested caffeine, making it more difficult to induce expectancies with the administration of a placebo. Overall, these results suggest that resistance-trained individuals familiarized with the use of caffeine-containing products do not obtain the placebo effect of caffeine on exercise performance. However, investigations including samples of individuals habituated to caffeine vs. individuals with low habitual caffeine intake, undergoing the same deceptive protocol to induce the placebo effect of caffeine, are necessary to undoubtedly confirm that habituation to caffeine may affect the placebo effect of caffeine on exercise performance. In these experiments, the measurements of beliefs about caffeine effectiveness in improving performance in both groups are required to determine the influence of expectancy on the placebo effect of caffeine. 

The lack of placebo effect of caffeine in individuals habituated to caffeine may be reinforced by outcomes of previous investigations. While the placebo effect of caffeine has been confirmed by using several deceptive protocols [10,11,14,15,16,17,18,19,20], the two investigations carried out with participants with a certain habituation to caffeine showed that the deception was not successful. Foad et al. [16] found that in participants ingesting at least 300 mg/day of caffeine, informing them that they had received caffeine when they had ingested placebo was not effective at increasing performance during a 40-km cycling time trial. In fact, the actual intake of 5 mg/kg of caffeine was effective to increase cycling performance, irrespective of whether they were informed that they received caffeine or placebo. Similarly, Tallis et al. [22] observed that the placebo effect of caffeine was not present in individuals with low caffeine consumption (92 mg/day) during a test of maximal voluntary concentric force of the knee flexors and extensors. Again, the actual ingestion of 5 mg/kg of caffeine was necessary to obtain muscle performance benefits in comparison to a control situation. All this information together points towards the lack of placebo effect of caffeine in endurance and resistance-based exercise in individuals who are habituated to caffeine. Thus, the acute ingestion of caffeine seems necessary to obtain the potential effect of this substance in habitual consumers of caffeine.

Despite the uniqueness of the presented results, there were several limitations in the experimental design employed which should be addressed to understand the significance of the outcomes. First, this study did not contain a double-dissociation design. This is because we did not aim to study the additive effect of being informed of receiving caffeine plus actually receiving caffeine, as previous investigations have done [19]. Instead, the purpose of the current investigation was to determine in isolation the placebo effect of caffeine in recreationally trained women habituated to this substance. For this reason, we used a deceptive experimental design that included two identical trials that differed only in the participant’s belief of having received caffeine. Second, the study did not include any measurement to assess how effective our deceptive protocol was in terms of participants’ actual belief of receiving caffeine and their expectancies regarding the positive effects of caffeine. Thus, the current investigation is unable to determine the effect of participants’ expectations on the placebo effect of caffeine in our study sample, and future investigation on this topic should include measurement of expectancy in samples of individuals habituated to caffeine. Third, the hormonal changes as a result of the menstrual cycle were not controlled in the investigation. Lastly, as our participants were mild–moderate caffeine users, it would also be interesting to assess withdrawal symptoms in the two trials to determine whether our participants perceived the lack of feelings habitually associated to caffeine. Despite these limitations, we believe the current manuscript is sound to determine the lack of the placebo effect of caffeine on recreationally trained women habituated to caffeine. The manuscript adds valuable information for resistance-trained women seeking to obtain benefits related to caffeine supplementation.

## 5. Conclusions

The results of the present study indicate that intake of a placebo in recreationally trained women with mild–moderate consumption of caffeine, while they are informed that they are ingesting 6 mg/kg of caffeine, is not an effective strategy to improve maximal strength as well as strength-endurance performance during the bench press exercise. This suggests the lack of the placebo effect of caffeine in individuals habituated to caffeine. To determine if habituation to caffeine modifies the potential psychological benefits obtained through a deceptive protocol to induce a placebo effect, future investigations should include a comparison of individuals with different levels of habituation to caffeine intake undergoing the same deceptive protocol to induce the placebo effect of caffeine.

## Figures and Tables

**Table 1 nutrients-12-03813-t001:** Participants’ characteristics.

Age (years)	23.0 ± 0.8
Body mass (kg)	62.5 ± 6.9
Height (cm)	167.5 ± 5.1
Resistance training experience (years)	3.0 ± 0.8
1 RM in bench press exercise (kg)	40.0 ± 9.7
Habitual caffeine intake (mg/b.m; mg/day)	4.1 ± 1.6; 248.8 ± 91.5
Energy intake (kcal)	2332.8 ± 255.1
Protein (% of total energy intake)	22.5 ± 2.8
Carbohydrates (% of total energy intake)	27.3 ± 3.7
Fat (% of total energy intake)	50.2 ± 4.8

**Table 2 nutrients-12-03813-t002:** Muscle performance variables for placebo (PLAC) and control (CONT) conditions.

Variable	CONT (95% CI)	PLAC (95% CI)	*p*	*d*	Relative Effect (%)
1RM (kg)	40.0 ± 10.5 (33.7 to 46.3)	41.0 ± 9.5 (35.2 to 46.7)	0.10	0.10	3.1 ± 4.9
SET repetitions (*n*)	32.2 ± 5.1 (29.2 to 35.3)	31.8 ± 4.5 (29.1 to 34.5)	0.66	0.08	−0.6 ± 11.5
SET time under tension (s)	48.2 ± 6.3 (44.4 to 52.1)	49.2 ± 7.4 (44.7 to 53.6)	0.67	0.15	2.8 ± 17.1
SET mean power output (W)	130 ± 34 (110 to 151)	121 ± 26 (106 to 137)	0.08	0.30	−5.4 ± 11.2
SET peak power output (W)	293 ± 82 (244 to 343)	268 ± 48 (239 to 297)	0.12	0.37	−5.5 ± 15.4
SET mean velocity (m/s)	0.62 ± 0.09 (0.56 to 0.67)	0.60 ± 0.10 (0.54 to 0.65)	0.43	0.21	−2.3 ± 14.8
SET peak velocity (m/s)	1.17 ± 0.10 (1.11 to 1.23)	1.11 ± 0.10 (1.06 to 1.17)	0.07	0.60	−4.5 ± 8.3

All data are presented as mean ± standard deviation and 95% confidence intervals for all participants. For the variables time under tension, mean power output, peak power output, mean velocity, and peak velocity, an average of the values obtained in all the repetitions performed in the strength-endurance test was calculated. 1RM = one-repetition maximum.

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
