# Peer review of "Placebo Effect of Caffeine on Maximal Strength and Strength Endurance in Healthy Recreationally Trained Women Habituated to Caffeine"

_nutrients, 2020, doi:10.3390/nu12123813_

Round 1

Reviewer 1 Report

It is a very interesting and ambitious study with an original design. Really intended to evaluate the placebo effect of caffeine in women is innovative.

Special mention to the introduction: Very good, extensive and well documented documented

I think there are some points that could improve the study:

2.1. Experimental design

Line 92: Is the phrase "ingestion of a capsule containing (PLAC) ..." unfinished?

2.2. Study participants

Perhaps the classic formula of "inclusion criteria / exclusion criteria" should be used. Thus, for example, it would be clear if the participants were taking some type of sports drug or supplementation that could alter the results; menstrual cycle (this is not explained until discussion); contraceptive use of all of them, some or none; exclusion of celiac women (I imagine that the PLAC flour was wheat), etc.

2.3. Familiarization session and one repetition maximum test

Are they learning, habituation, familiarization sessions? The need for it should be justified.

Lines 130-132: In order to replicate the study in another laboratory, the information provided to the participants should be more detailed.

2.4. Experimental protocol

In order to be able to replicate this protocol in another laboratory, some additional data would be useful: rest time between the approximation series; rest time between approximation series and RM; if the quality of the repetitions were evaluated or if they were all counted as correct.

3. Results

Table 1. I have doubts about the results of SET Mean velocity [m / s] and SET Peak velocity [m / s]

For example (*):

The lifting speed at 1RM in BP is usually 0.15m / s. Moreover, 1RM in BP is usually 0.94m / s.

Is it possible that all repetitions were averaged regardless of the test?

If so, it must be indicated. Alternatively all means of all parameters could be displayed.

* González-Badillo JJ, Sánchez-Medina L. Movement velocity as a measure of loading intensity in resistance training. Int J Sports Med. 2010 May;31(5):347-52.

4. Discussion

I have doubts whether explaining the effects of caffeine to the participants leads to a specific placebo effect of caffeine or energizing / anti-fatigue substances in general.

6. Conclusions

This study has not been given caffeine or evaluated action so should not give this advice: Instead, the actual ingestion of caffeine should be recommended [14,20] although high doses may be necessary to obtain the potential ergogenic effect of caffeine in athletes habituated to this substance [23,24]

Author Response

Response to Reviewer's comments

Placebo Effect of Caffeine on Maximal Strength and Strength-endurance in Healthy Recreationally Trained Women Habituated to Caffeine

We sincerely thank the Reviewers and the Nutrients Editors for careful peer-reviewing the manuscript and for their valuable comments provided in the review letter. Below, we have included a response letter where we have replied item-by-item to the comments provided by the Reviewers. We have highlighted the changes within the manuscript in yellow. We feel that the manuscript is improved in the light of the suggested changes.

Reviewer: 1
It is a very interesting and ambitious study with an original design. Really intended to evaluate the placebo effect of caffeine in women is innovative.

Special mention to the introduction: Very good, extensive and well documented

I think there are some points that could improve the study.

Reply - Thank you for this summary of our investigation and for the comments provided in this review letter.

 Comments:

 2.1. Experimental design

Line 92: Is the phrase "ingestion of a capsule containing (PLAC) ..." unfinished?

 Reply - Thank you for this comment, this sentence was re-written.

 Line 104:… (1) ingestion of a capsule containing a placebo (PLAC)

2.2. Study participants

Perhaps the classic formula of "inclusion criteria / exclusion criteria" should be used. Thus, for example, it would be clear if the participants were taking some type of sports drug or supplementation that could alter the results; menstrual cycle (this is not explained until discussion); contraceptive use of all of them, some or none; exclusion of celiac women (I imagine that the PLAC flour was wheat), etc.

Reply - Information about the inclusion criteria as well as information on other drugs or supplements has been added. Of note, women taking oral contraceptives were not discarded from the investigation as they would not have to ingest caffeine. Hence, interference of oral contraceptives on caffeine pharmacokinetics was not present.

Lines 122- 129: The inclusion criteria were as follows: (a) free from neuromuscular and musculoskeletal disorders, (b) habitual caffeine user within the range of mild/moderate consumption, as per previous proposed thresholds for classifying individuals in sport performance research according to their habitual caffeine consumption [Filip et al. 2020], (c) resistance trained, (d) no medication nor dietary supplements usage within the 3 previous months (oral contraceptives were allowed), (e) self-described satisfactory health status. Eleven women were regularly taking oral contraceptives for at least the three months prior to the investigation.  Participants were excluded if they reported a) a positive smoking status; b) potential allergy to caffeine. 

2.3. Familiarization session and one repetition maximum test

Are they learning, habituation, familiarization sessions? The need for it should be justified.

Lines 142- 145: Since the 1RM test and the sets until muscular failure were not an usual practice for participants, three weeks before the onset of the experimental trials, two familiarization sessions separated by one-week were performed, in order to restrict learning effects during the experiment and decrease the risk of injury [Grgic et al. 2020].

Lines 130-132: In order to replicate the study in another laboratory, the information provided to the participants should be more detailed.

Reply - We have included this information  

Lines 149-155: In the last familiarization session, participants were informed about the potential ergogenic effects of acute caffeine intake on maximal strength and on muscular strength-endurance. The expectancy of positive effects of caffeine on performance was induced by informal but structured and normalized conversations with participants provided by a qualified sport dietitian. Specifically, participants were shown a published infographic into caffeine and resistance exercise [Baltazar- Martins et al. 2019], and the dietitian analyzed benefits for sport and exercise performance and explained the guidelines for caffeine supplementation.

2.4. Experimental protocol

In order to be able to replicate this protocol in another laboratory, some additional data would be useful: rest time between the approximation series; rest time between approximation series and RM; if the quality of the repetitions were evaluated or if they were all counted as correct.

 Reply: As suggested, more details about the experimental sessions have been added.

 Lines 158-168: Then, participants performed a warm-up that included 10 min of cycling and a single set of 8 reps of 50% 1RM, 6 reps of 70% 1RM and 3 reps of 80% 1RM, established during the familiarization. The resting interval between the sets used for warming-up was of 3 minutes. Then, participants performed the 1RM test to assess upper-body maximal muscle strength. The initial load was set at 90% of the estimated 1RM and, in each subsequent attempt, the load was increased between 1.25 and 5 kg until the load that represented the 1RM was reached. The 1RM load was determined using a maximum of 2 to 5 attempts with a resting interval of 5 min between attempts. After five minutes of resting, the strength-endurance test was performed until muscular failure with the 50% of 1RM load, measured in the previous test. Muscular failure was defined as the inability to perform another concentric movement in its entire range of motion [Izquierdo et al., 2006]

 Lines 173-175: Execution technique during the strength-endurance test was monitored by two experienced researchers who gave a standardized verbal encouragement to aid in the obtaining of the highest velocity in each repetition.

  1. Results

 Table 1. I have doubts about the results of SET Mean velocity [m / s] and SET Peak velocity [m / s]. For example (*):

The lifting speed at 1RM in BP is usually 0.15m / s. Moreover, 1RM in BP is usually 0.94m / s. Is it possible that all repetitions were averaged regardless of the test? If so, it must be indicated. Alternatively all means of all parameters could be displayed.

Reply: We agree with the Reviewer, however, we analyzed the kinematic variables only in the strength-endurance test, and we do not have data of velocity in the 1RM test. The mean bar velocity obtained at 50%1RM is lower than usually obtained in previous investigation because the mean bar velocity presented in Table 1 is a value obtained from averaging all the repetitions performed in the set. We provided new information in the footnote of the Table 1.

 For the variables time under tension, mean power output, peak power output, mean velocity and peak velocity, an average of the values obtained in all the repetitions performed in the strength-endurance test was calculated.

  1. Discussion

I have doubts whether explaining the effects of caffeine to the participants leads to a specific placebo effect of caffeine or energizing / anti-fatigue substances in general.

Reply – Thank you for this comment.  In the case of caffeine, not all previous investigations on the placebo effect of caffeine use an explanation of the ergogenic effects of caffeine, mainly because the ergogenic effects of caffeine are well known and most people have experienced this substance.  However, to assure that participants were aware of the potential effects of caffeine on resistance-based exercise, we used an information explanation session,  based on previous experimental protocols that also manipulated the expectations of the group of participants (caffeine literature, presentations, verbal information) [Beedie et al. 2006, Foad et al. 2008, Pollo et al. 2008, Duncan et al. 2009]. 

  1. Conclusions

 This study has not been given caffeine or evaluated action so should not give this advice: Instead, the actual ingestion of caffeine should be recommended [14,20] although high doses may be necessary to obtain the potential ergogenic effect of caffeine in athletes habituated to this substance [23,24]

 Reply - Thank you for this comment.  The expert Reviewer is right. We have excluded this sentence.

 On behalf of all authors, thank you for the thorough and constructive review.

Reviewer 2 Report

The manuscript presented examines the effect of caffeine expectancy on measures of muscular strength and strength endurance. The manuscript is generally well presented and to a good standard or written English. Whilst the novelty of the work is not well placed, the results are not confirmatory of previous published work and as such, this data set provides an interesting addition to this areas of investigation. Please see below a number of comments/suggestions that should be considered in a revised version of the manuscript.   

The study is presented in a manner that infers habitual users may respond differently to non- or low caffeine consumers. However, it is not possible to accurately answer this question without the inclusion of a low or non-user group. There is a real opportunity to add novelty to this area of investigation with the inclusion of this additional data. If this is not possible, the direction of the communication needs reworking to tone down the focus on habitation, particularly in the development of the study rationale.

The rationale for habituation influencing expectancy is not robust. A counter argument might be the idea that caffeine consumption is high in this population due to a belief it will be beneficial? Hence, you have a high expectancy group? Although this argument and the interpretation of the data, would be more robust if expectancy was measured in this group.

As per the specific suggestion below, I have some concerns about the approach adopted to induce expectancy. This is a very important aspect of the study that has either been not well reported or not well thought-out.

There is a need to clarify at various points in the manuscript that the population examined are high habitual caffeine consumers, similar studies have examined this question in participants that are low habitual caffeine users.

Abstract

Line 26: not sure I agree, many use those that are low habitual caffeine consumers or do not report this information

Should the aim not reflect high habitual caffeine users?

Line 38: The first line of the conclusion needs to be updated to be more accurate as induced expectancy was not measured.  

Introduction

Line 47: Please provide a reference.  That in the later sentence does not support the statement regarding caffeine and training

Line 53: ‘a burst of research’ odd terminology consider rephrasing

Line 60: can this conclusion be drawn based on the evidence from lab based studies where double blind methods are used? This statement should be refocused to indicate that this may be an important mechanism when caffeine is consumed in sporting settings

Line 68: It would be worth including some critique of these protocols in relation to the results and to better frame the approach used in the present study. I feel it also important to highlight the ambiguity with respect to how expectancy is induced and measured in previous work.

Method

Line 113: so it is likely that at least some of the individuals would have been similar in there habitual caffeine use to other expectancy studies (https://www.ncbi.nlm.nih.gov/pmc/articles/PMC6212857/). Is this not misleading given the study rationale?

Were participants not asked to abstain from caffeine and intense physical activity prior to completing the exercise assessments?    

Line 125: it is not clear why you would want to restrict possible learning effects in a familiarisation trial?

Line 130: Given that expectancy effect is largely based on the manipulation aspect of such studies, this seems a little vague. What specifically were participants told? How was this delivered? Who was it delivered by? Was the approach standardised? Why would you discuss side effects is you are trying to induce a positive response? How do you know your approach was effective to induce expectancy?  

Statistical analysis – what data checks were performed to inform the decision to use parametric statistical analysis? A reference should be provided for the threshold values cited for effect size measures

Discussion

163: Or more specifically the protocol used in the present study

166: Is this based on direct evidence of suggestions in published work?

Line 177 onwards: As per the comments above, it is difficult to justify this without having measure expectancy. The method of delivering the information is arguably more important

Line 198: Are the results by Tallis et al. not further confirmation that habituation is not the most important factor influencing the results?

Line 205: Are the results unique based on the statements made prior to this?

Line 216: I am sure they are referred to as high caffeine users in other parts of the manuscript?

Author Response

Response to Reviewers’ comments

Placebo Effect of Caffeine on Maximal Strength and Strength-endurance in Healthy Recreationally Trained Women Habituated to Caffeine

 We sincerely thank the Reviewers and the Nutrients Editors for careful peer-reviewing the manuscript and for their valuable comments provided in the review letter. Below, we have included a response letter where we have replied item-by-item to the comments provided by the Reviewers. We have highlighted the changes within the manuscript in yellow. We feel that the manuscript is improved in the light of the suggested changes.

Reviewer: 2

 We sincerely thank the Reviewers and the Nutrients Editors for careful peer-reviewing the manuscript and for their valuable comments provided in the review letter. Below, we have included a response letter where we have replied item-by-item to the comments provided by the Reviewers. We have highlighted the changes within the manuscript in yellow. We feel that the manuscript is improved in the light of the suggested changes.

 The manuscript presented examines the effect of caffeine expectancy on measures of muscular strength and strength endurance. The manuscript is generally well presented and to a good standard or written English. Whilst the novelty of the work is not well placed, the results are not confirmatory of previous published work and as such, this data set provides an interesting addition to this areas of investigation. Please see below a number of comments/suggestions that should be considered in a revised version of the manuscript. 

Reply- Thank you for this positive comment and for the suggestions made in this letter.

The study is presented in a manner that infers habitual users may respond differently to non- or low caffeine consumers. However, it is not possible to accurately answer this question without the inclusion of a low or non-user group. There is a real opportunity to add novelty to this area of investigation with the inclusion of this additional data. If this is not possible, the direction of the communication needs reworking to tone down the focus on habitation, particularly in the development of the study rationale.

Reply- Thank you for insightful comment. We set this experiment to investigate the placebo effect of caffeine in a group of habitual caffeine consumers.  For this reason, we do not have data on an equivalent group of women with low habituation to caffeine.  The expert Reviewer is right indicating that comparing the placebo effect of caffeine to previous investigations that examined the placebo effect of caffeine in groups of low caffeine users does not completely determines that habituation to caffeine diminishes the placebo effect of caffeine.  Following this comment, we have changed several parts of the discussion to town down the inferences made regarding habituation from the current experiment.

Lines 227- 233: However, investigations including samples of individuals habituated to caffeine vs individuals with low habitual caffeine intake, undergoing the same deceptive protocol to induce the placebo effect of caffeine, are necessary to undoubtedly confirm that habituation to caffeine may affect the placebo effect of caffeine on exercise performance.  In these experiments, the measurements of beliefs about caffeine effectiveness to improve performance in both groups is required to determine the influence of expectancy on the placebo effect of caffeine.

The rationale for habituation influencing expectancy is not robust. A counter argument might be the idea that caffeine consumption is high in this population due to a belief it will be beneficial? Hence, you have a high expectancy group? Although this argument and the interpretation of the data, would be more robust if expectancy was measured in this group.

 Again, thanks for this clever comment.  Unfortunately, we did not record data about the reasons for using caffeine in our sample of individuals and we did not assess the magnitude of expectancy.  However, we have included your point of view in the discussion.

Lines 221- 225: Other potential reason for the lack of placebo effect of caffeine in habitual users of caffeine may be associated to the higher expectancy of caffeine’ effectiveness to increase performance, as this may be one of the reasons why the became users of this stimulant.  To this regard, it is probable that habitual users of caffeine are more prone to detect when they have actually ingested caffeine, making more difficult to induce expectancies with the administration of a placebo. 

As per the specific suggestion below, I have some concerns about the approach adopted to induce expectancy. This is a very important aspect of the study that has either been not well reported or not well thought-out. There is a need to clarify at various points in the manuscript that the population examined are high habitual caffeine consumers, similar studies have examined this question in participants that are low habitual caffeine users.

Reply - Thank you for this insightful opinion and the provided comments. Our responses to specific comments are provided below.

 Abstract

Line 26: not sure I agree, many use those that are low habitual caffeine consumers or do not report this information

Reply - The Reviewer is right.  This sentence has been changed to show this view.

Lines 24-28: However, most of these investigations in the placebo effect of caffeine used individuals with low caffeine consumption or did not report habitual caffeine consumption, especially in studies analyzing resistance-based exercise. Hence, it is unknown if habitual caffeine consumers also benefit from the placebo effect of caffeine on exercise performance.

Should the aim not reflect high habitual caffeine users?

Reply – Thank you for this comment. We have recently published a Discussion in this same journal with criteria to define habitual caffeine intake (PMID: 32326386).  Based upon the proposed thresholds for classifying individuals in sport performance research according to their habitual caffeine consumption, our participants were in the range of moderate consumers.  We have included this in the abstract and in the methods and discussion.

Line 38: The first line of the conclusion needs to be updated to be more accurate as induced expectancy was not measured.

Reply – The expert Reviewer is right.  The first line of conclusions in abstract has been changed.

Line 39: Conclusion: Informing participants that they were given caffeine, when in fact they received a placebo, did not modify any performance variable measured in this investigation.

Introduction

Line 47: Please provide a reference. That in the later sentence does not support the statement regarding caffeine and training

Reply –We have included a reference to support this sentence.

Line 53: ‘a burst of research’ odd terminology consider rephrasing

Reply - The terminology has been changed. Thank you.

Line 53: number of research

Line 60: can this conclusion be drawn based on the evidence from lab based studies where double blind methods are used? This statement should be refocused to indicate that this may be an important mechanism when caffeine is consumed in sporting settings

Reply - as suggested, the sentence has been changed.

Line 60-61: However, several investigations have suggested that the belief of having ingested caffeine may also increase performance [9,10], suggesting that psychological factors may also contribute as a mechanism of action when caffeine is consumed in sporting settings.

Line 68: It would be worth including some critique of these protocols in relation to the results and to better frame the approach used in the present study. I feel it also important to highlight the ambiguity with respect to how expectancy is induced and measured in previous work.

Reply – Thanks for this comment. We have introduced how expectancy is induced in previous works. 

Lines 68- 78: In previous investigations on this topic, the deceptive designs employed entailed different approaches to induce expectancy of caffeine consumption.  In general, expectancy was induced by telling individuals that they had received an ergogenic dose of caffeine but administering an inert substance instead.  Habitually, the performance outcomes of this protocol are compared to a control situation with no substance administration or by using a placebo-controlled situation in which participants are told that they have received a placebo and in fact, receive a placebo.  However, the most complex deceptive protocols include four trials with combination of information to the participants about the received substances and changes in the substances administered (i.e., informed caffeine/received caffeine, informed caffeine/received placebo, informed placebo/received caffeine, and informed placebo/received placebo). 

Method

Line 113: so it is likely that at least some of the individuals would have been similar in there habitual caffeine use to other expectancy studies (https://www.ncbi.nlm.nih.gov/pmc/articles/PMC6212857/). Is this not misleading given the study rationale?

Reply - Thank you for this comment. The expert reviewer is right when indicating that some individuals of previous investigations may be considered as habitual caffeine users. For example, in the study of Saunders et al. 2017 (https://pubmed.ncbi.nlm.nih.gov/27882605/) the range of habitual consumption was from 2-to- 583 mg/per day.  Unfortunately, in those studies data are reported as mean samples and it is impossible to dissociate if the placebo effect was higher in individuals with lower habituation to caffeine. Moreover, being a habitual caffeine user was not an inclusion criterion in these studies.  Hence, the novelty of this investigation is still associated to the use of habitual caffeine users.

 We have included the suggested reference in the introduction to clearly depict the association between the placebo effect and expectancy.

 Lines 61-64: To this regard, the increase in performance produced just by the perception of consuming caffeine, without actually consuming it, is often referred as placebo effect of caffeine which is associated with expectancy of oral caffeine consumption, as it most cases it is perceived as positive to enhance performance [Shabir et al. 2018].

 Were participants not asked to abstain from caffeine and intense physical activity prior to completing the exercise assessments?

Reply – Thank for pointing this out.  We have missed the inclusion of such information in the manuscript. Now, we provide this information.

 Line 133- 137: The participants were instructed to maintain their usual training, hydration and dietary habits during the study period (with the exception of refraining from any source of dietary caffeine for 12 h prior to trials). In addition, participants registered their food intake using “MyFitnessPal” software [29] 24 hours before the testing procedures. Last, participants were asked to avoid strenuous exercise for the 24 hours prior to each trial.

 Line 125: it is not clear why you would want to restrict possible learning effects in a familiarisation trial?

 Reply: Thanks for this comment. Although the order of the trials was randomized, we used familiarization trials to avoid the interference of the learning effects to the testing on the results of this investigation.

 Line 130: Given that expectancy effect is largely based on the manipulation aspect of such studies, this seems a little vague. What specifically were participants told? How was this delivered? Who was it delivered by? Was the approach standardised? Why would you discuss side effects is you are trying to induce a positive response? How do you know your approach was effective to induce expectancy? 

Reply – Thanks for this comment.  We provided more specific information to the manuscript.

Lines 149-155: In the last familiarization session, participants were informed about the potential ergogenic effects of acute caffeine intake on maximal strength and on muscular strength-endurance.  The expectancy of positive effects of caffeine on performance was induced by informal but structured and normalized conversations with participants provided by a qualified sport dietitian. Specifically, participants were shown a published infographic into caffeine and resistance exercise [Baltazar- Martins et al. 2019], and dietitian analyzed benefits for sport and exercise performance and explained the guidelines for caffeine supplementation.

We have also included the lack of measurement of expectancy as a limitation of the investigation.

 Statistical analysis – what data checks were performed to inform the decision to use parametric statistical analysis? A reference should be provided for the threshold values cited for effect size measures

Reply: Thank you for this comment. The Shapiro-Wilk, test was used in order to verify the normality of the sample data. This information has been added in Statistical Analysis section as well as reference to effect size thresholds.

 Line 177: The Shapiro-Wilk, test was used in order to verify the normality of the sample data.

Line 180: “… and trivial (d <0.2) [Cohen, 2013]”

Discussion

163: Or more specifically the protocol used in the present study

Line 173: The lack of placebo effect of caffeine was evident on all measured variables during protocol used in the present study

Thank you for this comment. We have rewritten this sentence.

166: Is this based on direct evidence of suggestions in published work?

Thank you for this comment. We have removed this sentence.

Line 177 onwards: As per the comments above, it is difficult to justify this without having measure expectancy. The method of delivering the information is arguably more important

 Thank you for this comment. We have reworded this paragraph to include a clearer view of the novelty of this investigation.

 Lines: 221- 225: Other potential reason for the lack of placebo effect of caffeine in habitual users of caffeine may be associated to the higher expectancy of caffeine’ effectiveness to increase performance, as this may be one of the reasons why the became users of this stimulant.  To this regard, it is probable that habitual users of caffeine are more prone to detect when they have actually ingested caffeine, making more difficult to induce expectancies with the administration of a placebo. 

Line 198: Are the results by Tallis et al. not further confirmation that habituation is not the most important factor influencing the results?

Thank you for this comment. In this investigation, participants were low caffeine users. We have indicated this in that sentence.

Line 205: Are the results unique based on the statements made prior to this?

Thank you for this comment. To avoid misleading potential readers, we have related the novelty of this investigation to the measurement of the placebo effect on women habituated to caffeine during resistance-based exercise. 

Line 216: I am sure they are referred to as high caffeine users in other parts of the manuscript?

Reply - Thank you for this comment. In line 124 we classified them as mild-moderate caffeine users and we used this classification in manuscript.

 On behalf of all authors, thank you for the thorough and constructive review.

Reviewer 3 Report

Placebo Effect of Caffeine on Maximal Strength and Strength-endurance in Healthy Recreationally Trained Women Habituated to Caffeine

Methods

  • Were participants instructed to modify their normal exercise routines in any way? Were they allowed to continue exercising if they were already doing so habitually?
  • How was the 1RM BP determined for the test session where participants performed a maximal number of repetitions at 50% of 1RM? From the timeline given in section 2.3, it would seem that 1RM BP must have first been established before proceeding to perform the repetitions at 50%.
  • What was the rest period given between each warm-up set?
  • Please clarify the BP warm-up session: is it three sets of 8, 6, 3 and 1 each (3x8, 3x6, 3x3, 3x1)? That would be a total of 54 reps which is likely to adversely affect the 1RM test due to fatigue.
  • Five minutes of recovery between the 1RM test and the SET test seems inadequate, particularly given the heavy warm-up session. Is it not likely that the results of the 50% 1RM test is affected by the previous test?
  • For the SET test, what was the criteria for the final repetition? Was it based on the participant’s volitional failure? Or was there a certain velocity (or form such as lowering the bar to a certain point) that needed to be maintained in order for the repetition to count?
  • Was supplement/medication use documented? Particularly those known to have an ergogenic effect such as BCAAs, creatine, leucine, beta alanine, amphetamines, etc.
  • Ideally, comparisons should be made between women habituated to caffeine and women not habituated to caffeine within the same study. In other words, a control group composed of women not habituated to caffeine could provide much stronger evidence that there is indeed a difference in placebo effect to caffeine due to caffeine habituation. However, as it stands, the lack of an observed effect may simply be due to other variables such as low sample size (there are interesting trends noted despite having p-values slightly above the prescribed 0.05). At the very least, this should be noted as either a limitation or as a future investigation to confirm findings.

Results

  • The baseline information of the participants’ characteristics that is included in section 2.2 could be given in a new table instead of within the text. This would drastically improve readability. Key characteristics to include would be age, height, weight, BMI, years of resistance training, habitual caffeine intake, calorie intake, intake of macronutrients,
  • Were any other dietary factors (not just proportion of macronutrients) different between experimental sessions?

Minor:

  • Line 212: should say “belief” instead of “believe”
  • Line 228: should say “recommended” instead of “recommend”

Author Response

Response to Reviewers’ comments

Placebo Effect of Caffeine on Maximal Strength and Strength-endurance in Healthy Recreationally Trained Women Habituated to Caffeine

 We sincerely thank the Reviewers and the Nutrients Editors for careful peer-reviewing the manuscript and for their valuable comments provided in the review letter. Below, we have included a response letter where we have replied item-by-item to the comments provided by the Reviewers. We have highlighted the changes within the manuscript in yellow. We feel that the manuscript is improved in the light of the suggested changes.

Reviewer 3:

 We sincerely thank the Reviewers and the Nutrients Editors for careful peer-reviewing the manuscript and for their valuable comments provided in the review letter. Below, we have included a response letter where we have replied item-by-item to the comments provided by the Reviewers. We have highlighted the changes within the manuscript in yellow. We feel that the manuscript is improved in the light of the suggested changes.

 Were participants instructed to modify their normal exercise routines in any way? Were they allowed to continue exercising if they were already doing so habitually?

 Reply - Thank you for this comment. Yes, participants were allowed to continuous with their exercise routines to avoid any detraining effects, but they avoided strenuous exercise 24 h before testing.  We have included this information within the manuscript.

 Lines 136-137:Last, participants were asked to avoid strenuous exercise for the 24 hours prior to each trial.

 How was the 1RM BP determined for the test session where participants performed a maximal number of repetitions at 50% of 1RM? From the timeline given in section 2.3, it would seem that 1RM BP must have first been established before proceeding to perform the repetitions at 50%.

Reply - Thank you for this comment. The expert Reviewer is right. First, it was measured 1RM and then, the strength-endurance test was performed with 50% of the 1RM measured in the prior test.

What was the rest period given between each warm-up set?

Reply - Thank you for this comment. We have included this information within the text.

 Lines 161: The resting interval between the sets used for warming-up was of 3 minutes.

Please clarify the BP warm-up session: is it three sets of 8, 6, 3 and 1 each (3x8, 3x6, 3x3, 3x1)? That would be a total of 54 reps which is likely to adversely affect the 1RM test due to fatigue.

Reply - Thank you for this comment. The previous explanation of this section was unclear. Participants performed only one set with each load in the warm-up. We have clarified this in the new version of the manuscript.

 Lines 158-168: Then, participants performed a warm-up that included 10 min of cycling and a single set of 8 reps of 50% 1RM, 6 reps of 70% 1RM and 3 reps of 80% 1RM, established during the familiarization. The resting interval between the sets used for warming-up was of 3 minutes. Then, participants performed the 1RM test to assess upper-body maximal muscle strength. The initial load was set at 90% of the estimated 1RM and, in each subsequent attempt, the load was increased between 1.25 and 5 kg until the load that represented the 1RM was reached. The 1RM load was determined using a maximum of 2 to 5 attempts with a resting interval of 5 min between attempts.

 Five minutes of recovery between the 1RM test and the SET test seems inadequate, particularly given the heavy warm-up session. Is it not likely that the results of the 50% 1RM test is affected by the previous test?

Reply - Thank you for this comment. The warm-up only included a single sets with submaximal loads.  In the pilot trial, we checked that 5 min was enough time to produce the feeling of complete recovery in the participants.  For this reason, we used this time between tests. Please, note that 1RM was achieved with only 2-5 attempts.

For the SET test, what was the criteria for the final repetition? Was it based on the participant’s volitional failure? Or was there a certain velocity (or form such as lowering the bar to a certain point) that needed to be maintained in order for the repetition to count?

 Reply: According to the Reviewer suggestion the experimental protocol section has been significantly improved and more details has been provided.

 Lines 165-168: After five minutes of resting, the strength-endurance test was performed until muscular failure with the 50% of 1RM load, measured in the previous test. Muscular failure was defined as the inability to perform another concentric movement in its entire range of motion [Izquierdo et al., 2006].

Was supplement/medication use documented? Particularly those known to have an ergogenic effect such as BCAAs, creatine, leucine, beta alanine, amphetamines, etc

Reply – Yes, such information was documented. Participants who reported the use of any dietary supplement were excluded from the investigation.

 Lines 126-127: (d) no medication nor dietary supplements usage within the 3 previous months (oral contraceptives were allowed)

 Ideally, comparisons should be made between women habituated to caffeine and women not habituated to caffeine within the same study. In other words, a control group composed of women not habituated to caffeine could provide much stronger evidence that there is indeed a difference in placebo effect to caffeine due to caffeine habituation. However, as it stands, the lack of an observed effect may simply be due to other variables such as low sample size (there are interesting trends noted despite having p-values slightly above the prescribed 0.05). At the very least, this should be noted as either a limitation or as a future investigation to confirm findings.

 Thank you for this comment. We included your suggestions as a recommendation for future investigations.

 Lines 227-233: However, investigations including samples of individuals habituated to caffeine vs individuals with low habitual caffeine intake, undergoing the same deceptive protocol to induce the placebo effect of caffeine, are necessary to undoubtedly confirm that habituation to caffeine may affect the placebo effect of caffeine on exercise performance. 

Results

The baseline information of the participants’ characteristics that is included in section 2.2 could be given in a new table instead of within the text. This would drastically improve readability. Key characteristics to include would be age, height, weight, BMI, years of resistance training, habitual caffeine intake, calorie intake, intake of macronutrients,

Reply - We included in section 2.2 a new table with participants’ characteristics. Thanks for the suggestion.

Were any other dietary factors (not just proportion of macronutrients) different between experimental sessions?

Reply – The participants was asked to maintain their typical dietary and hydration habits. Reports from “MyFitnessPal” was checked by a sports nutritionist and did not show any significant changes or other factors, which could impact on the obtain results.

Minor:

Line 212: should say “belief” instead of “believe”

Line 228: should say “recommended” instead of “recommend”

Reply - Thank you for these comments. We changed the spelling in the manuscript.

 On behalf of all authors, thank you for the thorough and constructive review.

Round 2

Reviewer 2 Report

The reviewer would like to thank the authors for considering the comments and suggestions. Please consider the points below in the final version of the manuscript.

Line 26: This sentence need rewording to be more specific. It is not unknown if habitual caffeine consumers benefit from the placebo effect of caffeine

Line 53: This sentence needs rewording.

Author Response

Response to Reviewers’ comments

Placebo Effect of Caffeine on Maximal Strength and Strength-endurance in Healthy Recreationally Trained Women Habituated to Caffeine

 Reviewer: 2

The reviewer would like to thank the authors for considering the comments and suggestions. Please consider the points below in the final version of the manuscript.

Again, we sincerely thank the Reviewer for careful peer-reviewing the manuscript and for their valuable comments. Below, we have included a response letter where we have replied to the comments provided by the Reviewer. We have highlighted the changes within the manuscript in yellow.

Line 26: This sentence need rewording to be more specific. It is not unknown if habitual caffeine consumers benefit from the placebo effect of caffeine

Reply – Thank you.  This sentence has been changed.

Line 26-28: Hence, it is unknown if habitual caffeine consumers benefit from the placebo effect of caffeine on exercise performance

Line 53: This sentence needs rewording.

Reply – Thank you.  This sentence has been changed.

Line 53-55: In the last years, a number of studies have supported the performance-enhancing effects of caffeine across a wide range of sporting activities including aerobic exercise [4] anaerobic-like exercise [5], and resistance exercise [6–8].